# Strain-Induced Structural Phase Transitions in Epitaxial (001) BiCoO_3_ Films: A First-Principles Study

**DOI:** 10.3390/nano13162342

**Published:** 2023-08-15

**Authors:** Hao Tian, Shuqi Cui, Long Fu, Hongwei Zhang, Chenggang Li, Yingqi Cui, Aijie Mao

**Affiliations:** 1School of Physics and Electronic Engineering, Zhengzhou Normal University, Zhengzhou 450044, China; haotian@zznu.edu.cn (H.T.); fulong@zznu.edu.cn (L.F.); hwzhang@zznu.edu.cn (H.Z.); chenggangli@zznu.edu.cn (C.L.); 2School of General Education, Wuchang University of Technology, Wuhan 430223, China; hgning@wut.edu.cn; 3Institute of Atomic and Molecular Physics, Sichuan University, Chengdu 610065, China

**Keywords:** first-principles study, phase transition, polarization

## Abstract

We have simulated BiCoO${}_{3}$ films epitaxially grown along (001) direction with density functional theory computations. Leading candidates for the lowest-energy phases have been identified. The tensile strains induce magnetic phase transition in the ground state (P4mm symmetry) from a C-type antiferromagnetic order to a G-type order for the in-plane lattice parameter above 3.922 Å. The G-type antiferromagnetic order will be maintained with larger tensile strains; however, a continuous structural phase transition will occur, combining the ferroelectric and antiferrodistortive modes. In particular, the larger tensile strain allows an isostructural transition, the so-called Cowley’s ‘‘Type Zero’’ phase transitions, from Cc-(I) to Cc-(II), with a slight volume collapse. The orientation of ferroelectric polarization changes from the out-of-plane direction in the P4mm to the in-plane direction in the $Pmc2_{1}$ state under epitaxial tensile strain; meanwhile, the magnetic ordering temperature T${}_{N}$ can be strikingly affected by the variation of misfit strain.

## 1. Introduction

Multiferroic materials, which combine broken space inversion symmetry with time inversion symmetry, have become one of the fastest growing research topics [1]. Two or more ferroic orders can coexist in a single phase in such materials, such as ferroelectric and magnetic orders [2,3]. The ferroelectric polarization breaks the space inversion symmetry, while the magnetic order breaks the time inversion symmetry. The couplings between these orders facilitate conditions for controlling the magnetization by an electric field and/or the polarization by a magnetic field, which open up the technological applications in the field of spintronics, microwave filters, energy storage, and resistive switching devices [4,5,6].

BiCoO${}_{3}$ (BCO), isostructural with ferroelectric PbTiO${}_{3}$, possesses a rather large displacement of the Co${}^{3+}$ cation away from the center of the typical oxygen octahedra, resulting in forming a pyramid, whereas the pyramidal coordination can evolve into octahedral coordination under high pressure accompanied by the high-to-low spin state transition and structural phase transition [7,8,9]. Such structural characteristics lead to rich application research. BiCoO${}_{3}$-based ferroelectrics could possess a significantly negative thermal expansion (NTE), whereby the volume shrinks rather than expands as the temperature rises, providing an opportunity to control the overall thermal expansion of structural materials [10]. By optimizing the arrangement of ligand vacancies, BiCoO${}_{3}$ could enhance OER activity at a large current density with low overpotential, meeting the requirements of industrial water splitting [11]. BiCoO${}_{3}$ was also proposed as a low-cost and high-performance electrode material. The retention of 92.7% after 5000 cycles at 1 Ag${}^{-1}$ current density and an almost invariable specific capacity over different current density cycles were achieved with a 3D urchin-like BiCoO${}_{3}$ material as a supercapacitor, which is free of templates and surfactants [12]. The N$\overset{´}{e}$el temperature could be changed significantly from 44 K for x = 0 to 470 K for x = 1 by varying the composition in the solid solution (1−x)PbVO${}_{3}$−xBiCoO${}_{3}$[13]. Due to the bulk Rashba effect, BiCoO${}_{3}$ are particularly promising for energy storage via specific spin texture [14,15].

On the other hand, the properties of thin films could be prominently affected by epitaxial strain derived from lattice mismatch between a thin film and the underlying substrate [16]. In this paper, we perform first-principles calculations to simulate BiCoO${}_{3}$ films epitaxially grown along the (001) direction. Due to the coupling of AFD and polarization, the polarization tends to tilt away from the out-of-plane in a phase transition between the tetragonal P4mm phase and the monoclinic Cc phase and eventually tends to lie along the in-plane direction in the orthorhombic $Pmc2_{1}$ state under epitaxial tensile strain, resulting in an epitaxial strain-induced phase transition. It can also be demonstrated that the magnetic ordering temperature T${}_{N}$ can be significantly enhanced above 800 K in the Cc phase and $Pmc2_{1}$ phase under tensile strain.

## 2. Computational Methods

We conducted first-principles calculations by the Vienna Ab initio Simulation Package (VASP) with projector augmented wave (PAW) pseudopotentials based on the density functional theory (DFT) [17,18]. The GGA-PBE exchange–correlation functional was employed to describe the electron–ion interaction [19]. We chose a cutoff energy of 550 eV and a Monkhorst–Pack k-point mesh with 6 × 6 × 4 after carefully evaluating the convergence of the computed results with regard to the cutoff energy and the number of k points. Note that the choice of a 550 eV energy cutoff is higher than 1.3 times the maximum value of ENMAX for oxygen’s pseudopotential and the 6 × 6 × 4 k-point mesh for a $\sqrt{2}\times \sqrt{2}\times 2$ unit cell defined as below is also adequately precise, as demonstrated in numerous studies and our previous work [20,21,22,23,24]. In order to model epitaxial (001) BiCoO${}_{3}$ films, the lattice vectors of these 20-atom cells (i.e., $\sqrt{2}\times \sqrt{2}\times 2$ unit cells) are defined as
(1)$$a=a_{IP}\left(\hat{x}+\hat{y}\right)$$
(2)$$b=a_{IP}\left(\hat{x}-\hat{y}\right)$$
(3)$$c=a_{\mathrm{IP}}\left[\delta_{1}\hat{x}+\delta_{2}\hat{y}+\left(2+\delta_{3}\right)\hat{z}\right]$$
where $\hat{x}$, *y*, and $\hat{z}$ are the unit vectors lying the substrate pseudocubic [100], [010], and [001] directions, respectively. a and b are the in-plane lattice vectors, while c is the out-of-plane lattice vector, which are along the pseudocubic [110], [$\bar{1}$10], and [001] directions, respectively. $a_{IP}$ is the in-plane lattice parameter corresponding to the epitaxial strain. Hence, the epitaxial strain can be obtained by $\eta =a_{IP}/a_{eq}-1$, where $a_{eq}$ = 3.72 Å corresponds to the in-plane lattice parameter of the equilibrium P4mm state of bulk BCO. The model of epitaxial structure grown on a cubic (001)-oriented substrate is shown in Figure 1a. Structural optimizations are carried out by relaxing the cell parameter $\delta_{1}$, $\delta_{2}$, and $\delta_{3}$ as well as the atomic positions for total energy minimization until the Hellmann–Feynman force of 0.005 eV Å${}^{-1}$ is reached. The strategy for structural optimizations can result in fixing the in-plane lattice parameters while optimizing the out-of-plane lattice parameter. The validity of such a strategy for modeling epitaxial films has been demonstrated in other systems, such as BiFeO${}_{3}$, SrZrO${}_{3}$, and NaNbO${}_{3}$ [21,25,26].

The ferroelectric polarization is computed by means of Berry phase [27,28]. One can also evaluate the polarization from the product of the Born effective charges with the atomic displacements by the approximated expression [29]:(4)$$P=\frac{e}{V}\sum_{\alpha \beta }Z_{\alpha \beta }^{*i}u_{\beta }^{i}$$
in which $Z_{\alpha \beta }^{*i}$ and $u_{\beta }^{i}$ represent the Born effective charges (BECs) tensor element of atom *i* and the corresponding displacement of atom *i* along the direction β=x,y,z from the considered ferroelectric phase to its corresponding paraelectric phase, respectively. The indices α and β run over the three Cartesian axes and index *i* runs over all the atoms in the unit cell. The structural symmetry of strained epitaxial phases are determined by the use of the FINDSYM program [30].

## 3. Results and Discussion

### 3.1. Structures

The perovskite BiCoO${}_{3}$ adopts the non-centrosymmetric polar point group C4v with space group P4mm (the group number 99). As shown in Figure 1a, it yields the tetragonal structure. One can estimate possible low-symmetry distortion from the cubic ABO${}_{3}$ perovskite structure by the use of the typical tolerance factor [31],
(5)$$t=\left(r_{O}+r_{A}\right)/\sqrt{2}\left(r_{O}+r_{B}\right)$$
where $r_{A}$, $r_{B}$, and $r_{O}$ denote the ionic radii of A, B, and oxygen atoms, respectively. For the typical polar perovskite BaTiO${}_{3}$ and PbTiO${}_{3}$ with the non-centrosymmetric P4mm state, the tolerance factor is 1.061 (1.062) and 1.022 (1.019) by the bond valence parameter (Shannon ionic radii) [32,33], respectively. The tolerance factor of BiCoO${}_{3}$, however, is 0.974, obtained simply by the bond valence parameter. Such a fact (*t* < 1) is inconsistent with BaTiO${}_{3}$ and PbTiO${}_{3}$ and further implies that the oxygen octahedra tiltings will yield a lower symmetry, such as tetragonal, orthorhombic, rhombohedral, etc. Hence, it is legitimate to infer that BiCoO${}_{3}$ would exhibit more complex phases under external conditions, for example, epitaxial strain. However, before considering the strain engineering, let us focus on the ground state of BiCoO${}_{3}$.

Analogous to the typical ABO${}_{3}$ perovskite, the primitive cell of BCO embodies five atoms to form one formula unit. By the first-principles calculation, the in-lattice parameter in the tetragonal P4mm phase of BCO is a${}_{0}$ = 3.718 for the primitive cell with five atoms. Note that we utilize the $\sqrt{2}\times \sqrt{2}\times 2$ unit cell containing 20 atoms for all computations, as mentioned in the Computational Methods section. The use of these enlarged supercells is advantageous for modeling the antiferromagnetic ordering and optimizing the distorted structures. The result is well consistent with the previous experiments [7,34]. Bi atom is located in the vertex of the tetragonal structure with the Wycoff coordinate 1a (0, 0, 0). However, Co and O atoms deviate from the center of both the body and face of the cell. Co atom occupies the Wycoff coordinate 1b (0.5, 0.5, 0.430), while O1 and O2 atoms are located in 1b (0.5, 0.5, 0.863) and 2c ( 0.5, 0, 0.342). The off-center Co atom bonds with the five O atoms, leading to an oxygen pyramid of CoO5 rather than an octahedra.

We first calculate the evolution in the total energy with the in-plane lattice parameter for the low-energy states. The ground state is a P4mm symmetry with C-type antiferromagnetic order. As we can see, the tensile strain will induce a magnetic phase transition from the C-type antiferromagnetic order to G-type order for the in-plane lattice parameter above 3.922 Å. Then, the P4mm state will change into a monoclinic state, i.e., Cc symmetry, when subject to the in-plane lattice constaint from the matching plane with substrates. The structural phase transition occurs with $a_{IP}$ = 4.03 Å.(4.18,4.47), where the NdScO${}_{3}$, BaSnO${}_{3}$, BiScO${}_{3}$, etc., substrates [35] can provide such epitaxial tensile strains. Meanwhile, the spin orders will remain the G-type antiferromagnetic order as the strain continues to increase. This is even the case when the system undergoes another structural phase transition.

Let us now turn our attention to the phase transition from the Cc-(I) to Cc-(II) state being of an isosymmetric phase. Such transition is rather different with the isosymmetric phase transition of Cc to Cc’ in strained BiFeO${}_{3}$[36], i.e., the tetragonal-like T phase to the rhombohedral-like R phase. In striking contrast to the rhombohedral-like R phase of BFO (with a c/a ratio of 1.3 and FeO5 pyramids), the two Cc states in BiCoO${}_{3}$ both possess CoO6 octahedra with an $a^{-}a^{-}c^{-}$ oxygen octahedral tilting pattern. Hence, the two Cc states should both belong to the rhombohedral-like R phase. Note also that isosymmetric phase transition, the so-called Cowley’s ‘‘Type Zero’’ phase transition [37,38], can abruptly alter c/a lattice parameter ratios and/or unit cell volume, but not change the space group symmetry (inclusive of translational symmetry). As we can see, the symmetry of Cc (Cs4 point group with Schoenflies’ symbol) does not change when the phase transition occurs but with a change of c/a ratio from 0.99 to 0.96, as well as with a volume collapse of 2.9%, which are representative of a strong first-order transition. We also predict a tilt change of out-of-plane lattice vector, i.e., c axis, deflecting from the epitaxial perpendicular direction of in-plane, accompanied by a decrease in magnitude of 0.82 degrees, as shown in Figure 2. Note that the Cc-(II) phase can be matched well with the lattice constant of LaLuO${}_{3}$ substrate [16]. Moreover, the structural phases at a finite temperature may form with a smaller strain.

As the strain continues to increase, the $Pmc2_{1}$ phase appears. Without the volume collapses and the change of axial ratio in comparison with the previous phase transition, both volume and axial ratios change continuously at the boundary of the phase transition. Nevertheless, it is still a first-order phase transition, since the change in energy is shown in Figure 1. To facilitate an experimental validation of our theoretical predictions, Table 1 summarizes the calculated lattice parameters and unit cell angles obtained for the relaxed P4mm, Cc-(I), Cc-(II), and $Pmc2_{1}$ phases when the in-plane lattice parameters ***a*****${}_{IP}$** are equal to 3.72, 4.03, 4.18, and 4.47 Å, respectively, which are redefined by the FINDSYM program.

### 3.2. Polarization and Antiferrodistortive Vetors

Note that the orientation of ferroelectric polarization tends to tilt away from the out-of-plane in the transition of P4mm to Cc and eventually tends to lie along the in-plane direction in the $Pmc2_{1}$ state under epitaxial tensile strain, which is shown in Figure 3a. In the P4mm phase, although the out-plane of polarization decreases with the in-plane lattice parameter, the magnitude of polarization is still rather large, at above 160 $\mu {C/\mathrm{cm}}^{2}$, up to the boundary of phase transition. For this boundary, the total polarization reduces to 151 $\mu {C/\mathrm{cm}}^{2}$ in the Cc-I state. While the out-plane of polarization decreases to 45 $\mu {C/\mathrm{cm}}^{2}$, the in-plane polarization appears, which is about 102 $\mu {C/\mathrm{cm}}^{2}$ along *y*-axis.

At the same time, the pyramidal CoO5 in the P4mm phase disappears and oxygen octahedrons in the Cc state begin to form and rotate, which leads to the so-called antiferrodistortion. Here, we use the antiferrodistortive vector (AFD) to determine its magnitude and direction, in which its axis provides the direction of oxygen octahedra rotating, whereas its magnitude provides the angle of such a rotation. For example, the AFD vector of the Cc-I state for this boundary should be (7.7${}^{\circ }$,7.7${}^{\circ }$,7.8${}^{\circ }$), which is about along the [111] direction. Such an AFD vector also corresponds to an $a^{-}a^{-}a^{-}$ octahedra tilting pattern (indicated by Glazer’s notations [39]). With the strain increasing, a decreasing out-of-plane component of the AFD vector causes the polarization to decrease, resulting from their coupling. Similar conditions also occur in the Cc-II phase as well as the boundary between Cc-I and Cc-II phases. Whereas the out-of-plane polarization of Cc-II phase is larger than that of Cc-I phase, the in-plane polarization is smaller than that of Cc-I. When subjected to the $Pmc2_{1}$ state, a vanishing in-plane component of the AFD vector (i.e., $a^{0}a^{0}c^{+}$ tilting pattern) results in the enhancement of out-of-plane polarization and the vanishing of in-plane polarization. The in-plane polarization exceeds 85 $\mu {C/\mathrm{cm}}^{2}$. We note that Ref. [26] also reported a polar $Pmc2_{1}$ phase induced by strain, with an $a^{-}a^{-}c^{+}$ octahedral tilting pattern, which is similar to our calculated Ima2 structure in Figure 1b, featuring in-plane anti-phase rotations (which are higher in energy for our system, with the out-of-plane in-phase tilting suppressed during structural optimization). In contrast, the polar $Pmc2_{1}$ phase of BiCoO${}_{3}$ in this work has an $a^{0}a^{0}c^{+}$ tilting pattern, with the in-plane anti-phase tiltings suppressed and out-of-plane in-phase rotation activated, resulting in lower energy. Hence, the two structures are distinct. Furthermore, NaNbO${}_{3}$ has a $4d^{0}$ electronic configuration for Nb${}^{5+}$ without magnetism as a ferroelectric. BiCoO${}_{3}$ (with $3d^{6}$ for Co${}^{3+}$) exhibits antiferromagnetic ordering with a N$\overset{´}{e}$el temperature above room temperature as discussed later, which makes it a candidate for a room-temperature multiferroic material.

### 3.3. N$\overset{´}{e}$el Temperature

The N$\overset{´}{e}$el temperature (T${}_{N}$) can be estimated with the use of the mean field approximation (MFA) or Monte Carlo (MC) simulations [40,41] by considering the in-plane and out-of-plane magnetic exchange constants ($J_{\|}$ and $J_{⊥}$ ). According to the effective Heisenberg Hamiltonian model $H=\frac{1}{2}\sum_{<i,j>}J_{ij}S_{i}\cdot S_{j}$. The magnetic exchange constants can be determined from our first-principles calculations by mapping the calculated total energies for each magnetic state to the Heisenberg model:(6)$$\begin{array}{cc} J_{⊥} & =\left(1/4S^{2}\right)\left[E\left(F\right)-E\left(G\right)-E\left(A\right)+E\left(C\right)\right]\\ J_{\|} & =\left(1/8S^{2}\right)\left[E\left(F\right)-E\left(G\right)+E\left(A\right)-E\left(C\right)\right] \end{array}$$
where E(F), E(A), E(C), and E(G) are the total energies of the ferromagnetic (FM) and A-, C-, and G-type antiferromagnetic orders (A-AFM, C-AFM, and G-AFM), respectively. Note that the A-AFM order exhibits the spins of Co${}^{3+}$, which are parallel in the in-plane direction and antiparallel along the out-of-plane direction. The C-AFM order possesses the spins of Co${}^{3+}$, which are parallel along the out-of-plane direction and antiparallel in the in-plane direction. The G-AFM order yields the spins of Co${}^{3+}$ antiparallel to all the nearest neighbors. We assume normalized spins and take S=2 for simplicity here. The N$\overset{´}{e}$el temperature for the magnetic transition can then be evaluated in the MFA level as:(7)$$T_{N}∼\frac{2}{3}\left(2J_{\|}+J_{⊥}\right).$$

The calculated N$\overset{´}{e}$el temperature of bulk BiCoO${}_{3}$ perovskite oxide for different U${}_{eff}$ values are shown in Figure 4a. These results reveal that the transition temperature from the C-type antiferromagnetic state to the paramagnetic state of the P4mm phase for BiCoO${}_{3}$ is significantly influenced by the value of U${}_{eff}$. Note that the experimental value of T${}_{N}$, i.e., 474 K, was reported, which is marked by the gray and magenta dashed lines in the panels (**a**) and (**b**) and presents a direct comparison between our theoretical predictions and the measured transition temperatures. Hence, we employ the conventional value of 4.8 eV for U${}_{eff}$ with the Co ions. Such a value is consistent with the measured values, when recalling that mean field approaches neglect spin fluctuations and thus overestimate magnetic transition temperatures [23,42]. The corresponding local magnetic moment is about 3.10 $\mu_{B}$ for each Co ion, which is consistent with the +3 charge of this Co ion and is in rather good agreement with the measurements.

We also solve the Hamiltonians to obtain T${}_{N}$ by means of Monte Carlo simulations of a 20 × 20 × 20 periodically repeated simulation supercell that contains 8000 Co${}^{3+}$ spins. We use 40,000 Monte Carlo sweeps for thermalization and 10,000 additional sweeps for computing statistical averages. The specific heat can be calculated in MC simulations by
(8)$$C_{v}=\frac{\left(\langle E^{2}\rangle -{\langle E\rangle }^{2}\right)}{T^{2}}$$
once the system has reached equilibrium at the given temperature (T). Hence, T${}_{N}$ can be obtained by locating the maximum on the curve of $C_{v}$ versus T. We perform the MC modeling to determine the transition temperature of the P4mm phase in BiCoO${}_{3}$. As one can see, the specific heat exhibits a substantial reduction when heating over 410 K, indicating a transition from antiferromagnetic state to paramagnetic state at such a temperature. When recalling that the Monte Carlo simulations usually underestimate the magnetic ordering temperature, we thus employ the mean field approximation by fitting the value of U${}_{eff}$ to estimate the N$\overset{´}{e}$el temperature. Figure 4b depicts the transition temperature of the resulting phases possessing the lowest energy as a function of the in-plane lattice parameter. The blue and red data represent the transition temperature estimated by U${}_{eff}$ = 4.8 and 5.91 eV, respectively. Note that, with the modified U${}_{eff}$ = 4.8 eV, the magnetic order transition temperature exceeds 800 K via a strain-induced structural phase transition.

## 4. Conclusions

In summary, we have used density functional theory calculations to investigate misfit strain-induced structural phase transition in epitaxial (001) BiCoO${}_{3}$ films. Tensile strain induces a magnetic phase transition, that is, from a C-type antiferromagnetic order to a G-type order, without changes in spatial symmetry (P4mm space group). BiCoO${}_{3}$ film maintains the G-type antiferromagnetic order under larger tensile strains, but undergoes a continuous structural phase transition from P4mm to Cc to $Pmc2_{1}$ phase in the wake of couplings between the polarization and antiferrodistortive vectors. Strikingly, two isostructural Cc phases occur during the isostructural phase transition, resulting from the so-called Cowley’s ‘‘Type Zero’’ phase transitions. Under epitaxial tensile strain, ferroelectric polarization changes from out-of-plane to in-plane direction, whereas the magnetic ordering temperature T${}_{N}$ can be significantly enhanced above 800 K in the Cc phase and $Pmc2_{1}$ phase under tensile strain. In ultra-thin ferroelectric and multiferroic films subjected to large epitaxial strains, extrinsic factors (such as temperature, as well as materials’ defects, impurities, thicknesses, interfacial effects, and surface conditions) may impact the emergent phases and physical properties observed experimentally. Our findings exhibit a rich multiferroic phase and broaden its potential application based on multifunctional thin film devices.

## Figures and Tables

**Figure 1 nanomaterials-13-02342-f001:**
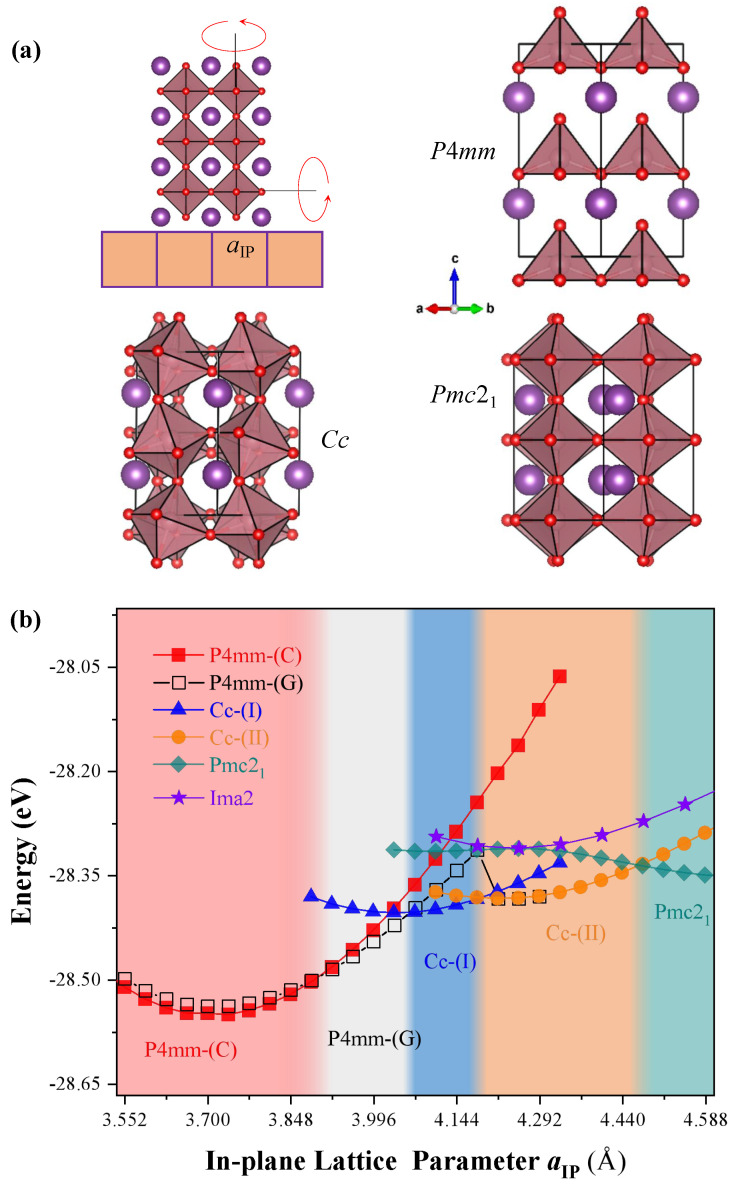
(**a**) The model of epitaxial structure grown on a cubic (001)-oriented substrate and the crystal structures for the lowest-energy phases, that is, P4mm, Cc, and $Pmc2_{1}$. The oxygen octahedra tilting in in-plane and out-of-plane directions (red oval arrows) can result in distorted structures with the antiferrodistortive vectors. (**b**) Total energy of the leading candidates for lowest-energy phase versus the in-plane lattice parameter in epitaxial (001) BCO films.

**Figure 2 nanomaterials-13-02342-f002:**
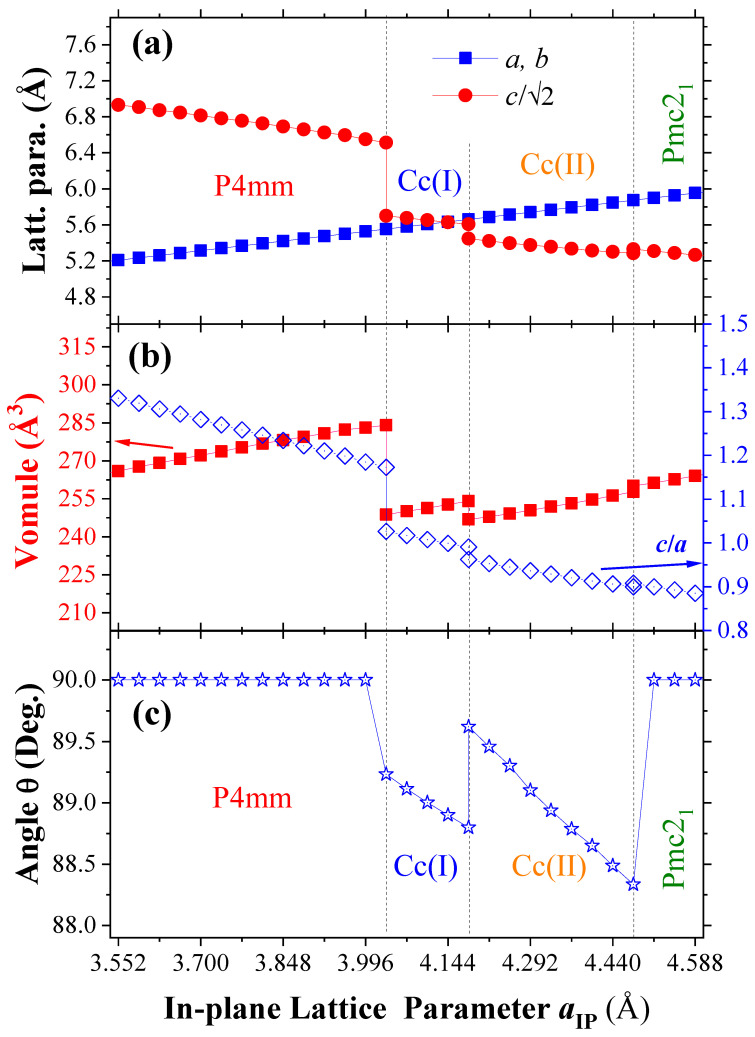
Structural properties of epitaxial (001) BCO films as a function of in-plane lattice parameter in the equilibrium phases. Panel (**a**) displays the lattice parameter *a*, *b*, and $c/\sqrt{2}$ versus in-plane lattice parameter. Panel (**b**) shows the volume (left vertical axis) and axial ratio (right vertical axis) versus the in-plane lattice parameter. Panel (**c**) represents the angle between the *c*-axis and the ab-plane versus in-plane lattice parameter.

**Figure 3 nanomaterials-13-02342-f003:**
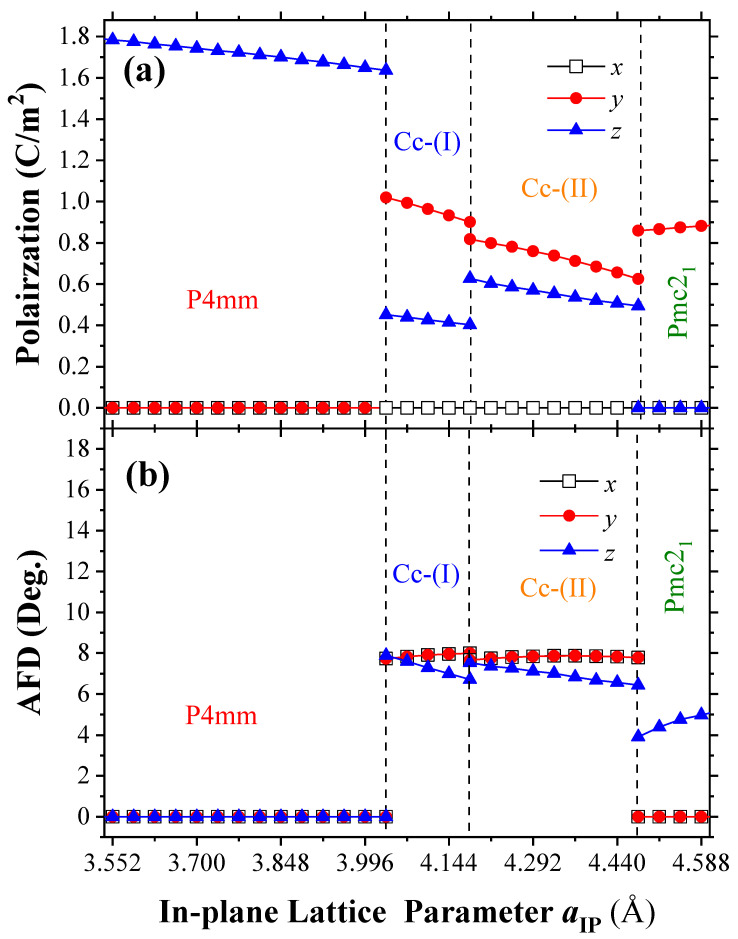
The Cartesian components of (**a**) the ferroelectric polarization and (**b**) antiferrodistortive (AFD) vectors of the equilibrium phases of epitaxial (001) BCO films, as a function of in-plane lattice parameter.

**Figure 4 nanomaterials-13-02342-f004:**
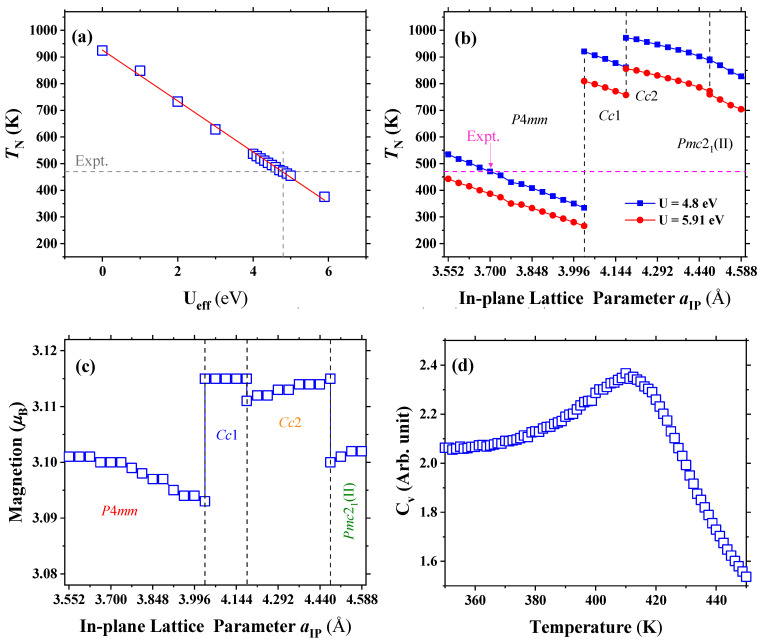
Magnetic-related properties of epitaxial (001) BCO films. Panel (**a**) displays the magnetic ordering temperature T${}_{N}$ varying with the Hubbard parameter U${}_{eff}$ for the ground state of BiCoO${}_{3}$. Panels (**b**,**c**) show the T${}_{N}$ with U${}_{eff}$ = 4.8 eV and 5.91 eV and magnetic moment of Co${}^{3+}$ as a function of in-plane lattice parameter. Panel (**d**) represents the temperature dependency of the specific heat $C_{v}$ in the ground state of BiCoO${}_{3}$. The gray and magenta dashed lines in the panels (**a**,**b**) denote the experimentally determined N$\overset{´}{e}$el temperatures (T${}_{N}$) of the tetragonal P4mm phase.

**Table 1 nanomaterials-13-02342-t001:** The reduced unit cell parameters of the P4mm, Cc-(I), Cc-(II), and $Pmc2_{1}$ phases when the in-plane lattice parameters ***a*****${}_{IP}$** are equal to 3.72, 4.03, 4.18, and 4.47 Å, respectively, which are redefined by the FINDSYM program. **a**, **b**, and **c** are the lattice parameters (in Å). **α**, **β**, and **γ** are reduced unit cell angles (in degrees). The number **n** is the number of atoms in the reduced unit cell of BiCoO${}_{3}$.

Phase	*a* ${}_{\mathrm{IP}}$	n	a	b	c	α	β	γ
P4mm	3.72	5	3.718	3.718	4.817	90	90	90
Cc-(I)	4.03	20	9.809	5.699	5.699	90	124.75	90
Cc-(II)	4.18	20	9.678	5.911	5.911	90	127.26	90
$Pmc2_{1}$	4.47	10	3.769	6.321	6.321	90	90	90

## Data Availability

The data are available on request from corresponding author.

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
