# Peer review of "Strain-Induced Structural Phase Transitions in Epitaxial (001) BiCoO3 Films: A First-Principles Study"

_nanomaterials, 2023, doi:10.3390/nano13162342_

Round 1

Reviewer 1 Report

The authors explore 2D material, which is an actual and modern area of research. The literature review is brief but mentions key works. The study of material properties under the action of deformations is important for the practical design of various sensors. Understanding phase transformations exhibit a rich multiferroic phase and broaden its potential application based on multifunctional thin film devices. This study is dedicated to the systematic study of BiCoO3 films. The authors use proven calculation schemes based on the density functional theory, so the results obtained must be correct. The figures presented in the work are clear and understandable. Thus, the authors investigated the properties of BiCoO3 film and found phase transitions.
1. As far as I understand, there are no experimental data.
2. It makes sense to describe the modeling procedure in more detail for a wide range of readers. What film thickness was simulated?
3. Add drawing of epitaxial (001) BiCoO3 film.
4. It makes sense for the authors to point out the differences from work (21) in which (001) NaNbO3 films under epitaxial strain are studied.

Reviewer 2 Report

Dear Editor,

I revised the manuscript entitled “Strain-induced structural phase transitions in epitaxial (001) BiCoO3 films: A first-principles study” by Hao Tian et al.

The Authors investigated misfit strain induced structural phase transition in epitaxial (001) BiCoO3 films by means of density functional theory calculations. The whole work is interesting, but there are some major concerns to address:

1-      In the introduction section the authors described inversions of symmetry, pyramidal and octahedral structures. I strongly encourage the authors to include a 3D structures picture, very helpful to understand these concepts, as reported in other papers with similar approaches.

2-      In the computational methods, the authors chose a cutoff energy of 550 eV and a Monkhorst-Pack k-point mesh with 6x6x4. 550 eV, which is a really low value: the authors should increase the cut-off, at least by doubling the actual value.

3-      The authors obtained the structural symmetry of strained epitaxial phases using the FINDSYM program. In the results section, the authors should include the lattice parameters which define the unit cell. Both angles and the axes values should be included: this is crucial to identify the correct unit cell for the simulations.

Round 2

Reviewer 2 Report

The Authors answered to all questions raised.

Author Response

Thanks for the comments "The Authors answered to all questions raised."